# A clustering-independent method for finding differentially expressed genes in single-cell transcriptome data

Alexis Vandenbon [1,2✉] & Diego Diez [3]

A common analysis of single-cell sequencing data includes clustering of cells and identifying differentially expressed genes (DEGs). How cell clusters are defined has important consequences for downstream analyses and the interpretation of results, but is often not straightforward. To address this difficulty, we present singleCellHaystack, a method that enables the prediction of DEGs without relying on explicit clustering of cells. Our method uses Kullback–Leibler divergence to find genes that are expressed in subsets of cells that are non-randomly positioned in a multidimensional space. Comparisons with existing DEG prediction approaches on artificial datasets show that singleCellHaystack has higher accuracy. We illustrate the usage of singleCellHaystack through applications on 136 real transcriptome datasets and a spatial transcriptomics dataset. We demonstrate that our method is a fast and accurate approach for DEG prediction in single-cell data. singleCellHaystack is implemented as an R package and is available from CRAN and GitHub.

[1] Institute for Frontier Life and Medical Sciences, Kyoto University, 53 Shougoin Kawara-cho, Sakyo-ku, Kyoto 606-8507, Japan. [2] Institute for Liberal Arts and Sciences, Kyoto University, Yoshidanihonmatsu-cho, Sakyo-ku, Kyoto 606-8501, Japan. [3] Immunology Frontier Research Center, Osaka University, 3-1 Yamada-oka, Suita, Osaka 565-0871, Japan. ✉email: alexisvdb@infront.kyoto-u.ac.jp

R ecent advances in single-cell technologies enable us to assess the state of cells by measuring different modalities like RNA and protein expression with single-cell resolution[1–5]. Hundreds of bioinformatics tools have been developed to process, analyze, and interpret the results from single-cell genomics data[6], such as monocle 2 and Seurat[7,8].

A standard protocol for analyzing single-cell data includes dimensionality reduction methods, such as principal component analysis (PCA), t-distributed stochastic neighbor embedding (t-SNE), and uniform manifold approximation and projection (UMAP) to visualize the data in fewer (typically 2) dimensions[9,10]. In addition, cells are often clustered, and differentially expressed genes (DEGs) are identified between the different clusters. This approach for finding DEGs by comparing between clusters is widely used in existing methods[8,11,12], and enables finding cluster-specific marker genes that facilitate labeling different cell populations. However, recent comparisons found that DEG prediction approaches for bulk RNA-seq do not generally perform worse than methods designed specifically for single-cell RNA-seq (scRNA-seq), and that agreement between existing methods is low[13,14]. Defining more flexible statistical frameworks for predicting complex patterns of differential expression is one of the grand challenges in single-cell data analysis[15].

A major problem with clustering-based approaches for DEG prediction is that the definition of cell clusters is often not straightforward. The number of biologically relevant clusters in a dataset is often not obvious. The high dimensionality of the data makes it hard to evaluate if the number of clusters and their borders make biological sense or if they are arbitrary. Furthermore, some cell sub-populations may not cluster independently and their defining signature may end up being obscured within a larger cluster. This can be critically important for low abundance populations in experiments using unsorted cells from tissue, where only a few representative cells may be present. Thus, the clustering of cells has important consequences in the interpretation of results and downstream analyses.

To address this problem we present singleCellHaystack, a methodology that uses Kullback–Leibler Divergence ($D_{KL}$; also called relative entropy) to find genes that are expressed in subsets of cells that are non-randomly positioned in a multidimensional (≥2D) space[16]. In our approach, the distribution in the input space of cells expressing or not expressing each gene is compared to a reference distribution of all cells. From this, the $D_{KL}$ of each gene is calculated and compared with randomized data to evaluate its significance. Thus, singleCellHaystack does not rely on clustering of cells, and can identify differentially expressed genes in an unbiased way. singleCellHaystack is implemented as an R package and is available from CRAN and GitHub.

## Results

### DEG prediction based on the expression distribution of genes.
We focus on the Tabula Muris bone marrow dataset to illustrate the principle of DEG prediction using the expression distribution of a gene (i.e., the distribution in the input space of cells that express a gene). Figure 1 shows t-SNE plots based on the 50 first principal components (PCs) of this dataset. Panels on the left show the detection of four genes (*Fcrla*—marker for B cells progenitors, *Fcnb*—marker for granulocytopoietic cells, *Lyz1*—marker for monocytes and granulocytes, *Ccl5*—marker for natural killer cells) in each cell. Each gene is expressed in a subset of cells occupying a subspace of the 50-dimensional input space. Genes with a non-uniform expression distribution are DEGs.

While estimating the distribution of cells in high-dimensional spaces is not straightforward, singleCellHaystack uses the local density of cells around several grid points as an approximation. Panels on the right in Fig. 1 show the grid points used by singleCellHaystack (100 by default; here coordinates of grid points in the 50 PC space were mapped to their approximate location in the t-SNE space), colored by the relative density of cells expressing each gene ($P(G = T)/Q$; see "Methods"). singleCellHaystack converts these distributions to $D_{KL}$ values and *p*-values reflecting the differential expression of each gene. As can be seen in Fig. 1, the four marker genes are DEGs (they are expressed only in subsets of cells) and they have a non-uniform expression distribution.

Our method contains two main functions: `haystack_highD` and `haystack_2D`, for multidimensional (≥2D) and two-dimensional (2D) input spaces, respectively. Input spaces could consist of principal components (PCs), t-SNE or UMAP coordinates, or the coordinates of cells in 2D or three-dimensional space for spatial transcriptomics data. The concept behind these two functions is the same (for an overview of the singleCellHaytack methodology we refer to the "Methods" section and Supplementary Fig. 1). Unless stated otherwise, results presented here were obtained using the `haystack_highD` function.

### Comparison with other methods using artificial datasets.
To evaluate the accuracy of our approach, we applied singleCellHaystack on 200 artificial datasets of varying size and complexity made using Splatter in which true DEGs are known (see "Methods")[17]. We also applied existing methods (DEsingle, EMDomics, scDD, edgeR, monocle 2 and approaches available through the `FindAllMarkers` function of the Seurat toolkit; see Supplementary Table 1) on the same datasets and compared their accuracy and runtimes[7,8,12,18–21]. The accuracy of each method was estimated using the area under the receiver operating characteristic (ROC) curve (AUC). To make a fair comparison, we applied Seurat's `FindAllMarkers` function both with and without its default filter, which is typically only passed by a small subset of genes.

Figure 2 shows the results for a selection of methods (see Supplementary Fig. 2 and Supplementary Tables 2 and 3 for all evaluated methods). Except for the smallest dataset size (1000 cells), singleCellHaystack shows comparatively high performance (Fig. 2a). In general, the accuracy of methods decreases with the complexity of the dataset, but the reduction was stronger for most of the cluster-based approaches (especially EMDomics, scDD, monocle 2, and the Wilcoxon rank-sum test without filter). One cause of the decreasing accuracy is the inaccurate clustering of cells in the larger datasets. Of note, scDD consistently failed to run successfully and returned errors on 2 and 9 of the datasets of 9000 cells and 10,000 cells, respectively.

Among the cluster-based methods, DEsingle and scDD have relatively high accuracy, but also have by far the longest runtimes (Fig. 2b and Supplementary Table 3). For example, median runtimes on 8000 cells for DEsingle and scDD were 123 and 81 h (!) respectively. In contrast, singleCellHaystack and the Wilcoxon rank-sum test implemented in Seurat (with default filter) are the fastest methods, taking about 5 min even on datasets of 10,000 cells. Removing the default filter of Seurat's Wilcoxon rank-sum test leads to higher accuracy (Fig. 2a) but longer runtimes (Fig. 2b).

These results show that singleCellHaystack has relatively high accuracy on artificial datasets and short runtimes, making it an attractive method for exploring single-cell datasets. Moreover, we stress that the artificial datasets made by Splatter are expected to give an advantage to cluster-based methods, since the underlying model of Splatter is based on generating clusters of cells. In

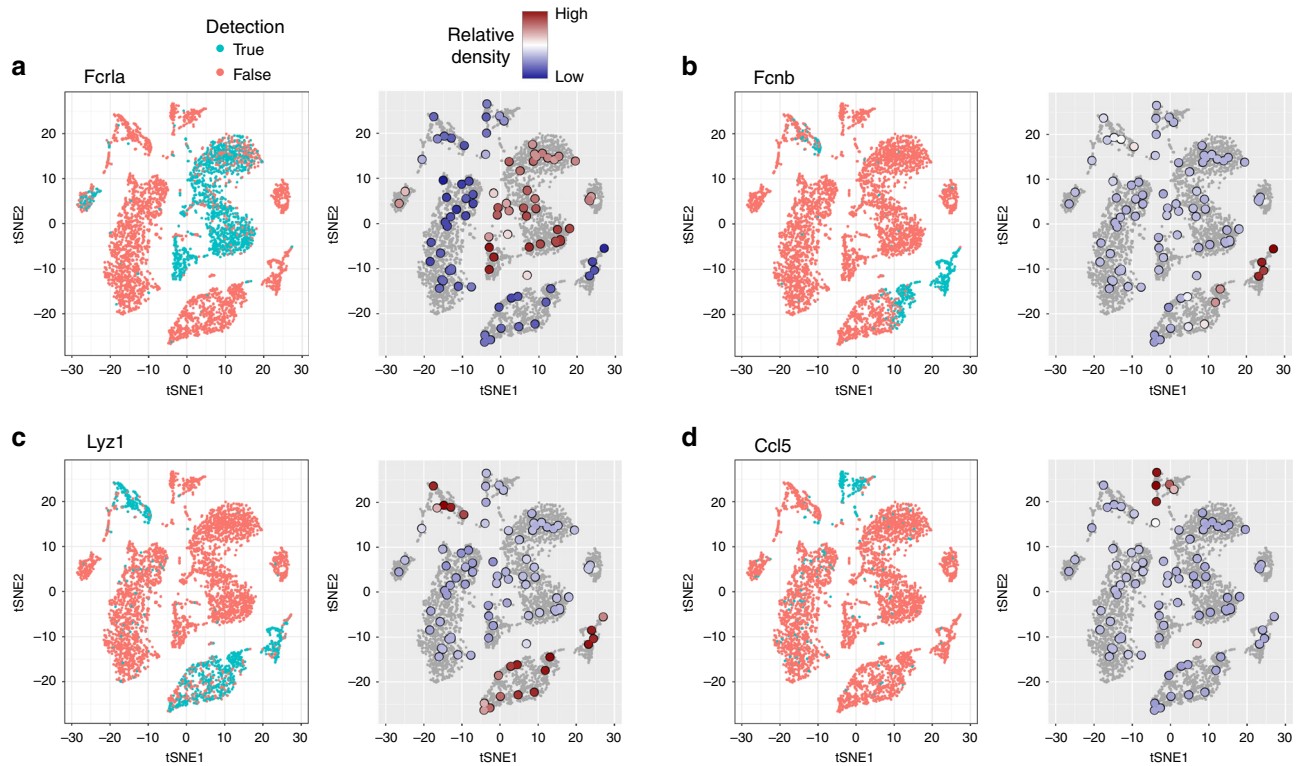

**Fig. 1 Illustration of the expression distribution approach for DEG prediction. a–d** For four example genes in the Tabula Muris bone marrow tissue dataset, t-SNE plots are shown, with the detection of the gene (left) and the expression distribution of the gene (right). In the expression distribution plots, colored circles represent the 100 grid points decided by singleCellHaystack, their color reflecting $P(G = T)/Q$. t-SNE coordinates of grid points are approximate: grid points were projected to the t-SNE coordinates of their most proximal cell in the 50 PC space.

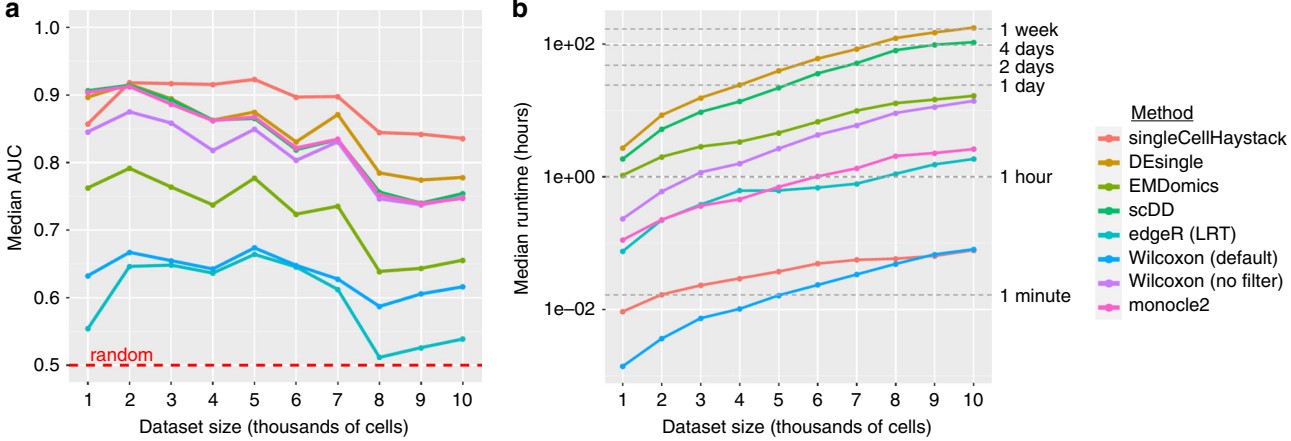

**Fig. 2 Comparison of DEG prediction methods applied on artificial datasets.** For a selection of DEG prediction methods the median AUC values (**a**) and median runtimes (**b**) are shown in function of dataset size. Medians are based on 20 datasets, except for scDD on datasets of size 9,000 (18 datasets) and 10,000 (11 datasets). In **a** the red dotted line shows the expected AUC for a random classifier. In **b** the gray dotted lines indicate 1 min, 1 h, 1, 2 and 4 days, and 1 week to improve the readability of the plot. Supplementary Fig. 2 shows similar plots for all evaluated methods.

addition, Splatter is based on a negative binomial model, giving an advantage to DEG prediction approaches that are based on a similar model. Indeed, DEsingle is based on a negative binomial model and performed comparatively well in our comparison. Nevertheless, singleCellHaystack had the highest performance, despite not being based on clusters nor on a negative binomial model.

**Application to 136 real single-cell datasets.** Next, we applied singleCellHaystack on 136 real scRNA-seq datasets of varying

sizes (149 to 19,693 cells). Median runtimes of `haystack_highD` using 50 PCs as input were 102 and 115 s using the simple and advanced mode (see "Methods" section), respectively. Runtimes followed an approximately linear function of the number of cells in each dataset (Supplementary Fig. 3A, B). Median runtimes for `haystack_2D` on 2D t-SNE coordinates were 75 and 84 s using the simple and advanced mode, respectively (Supplementary Fig. 3C, D).

In all datasets, large numbers of statistically significant DEGs were found. This observation is not surprising, since samples typically include a variety of different cell types. Rather than

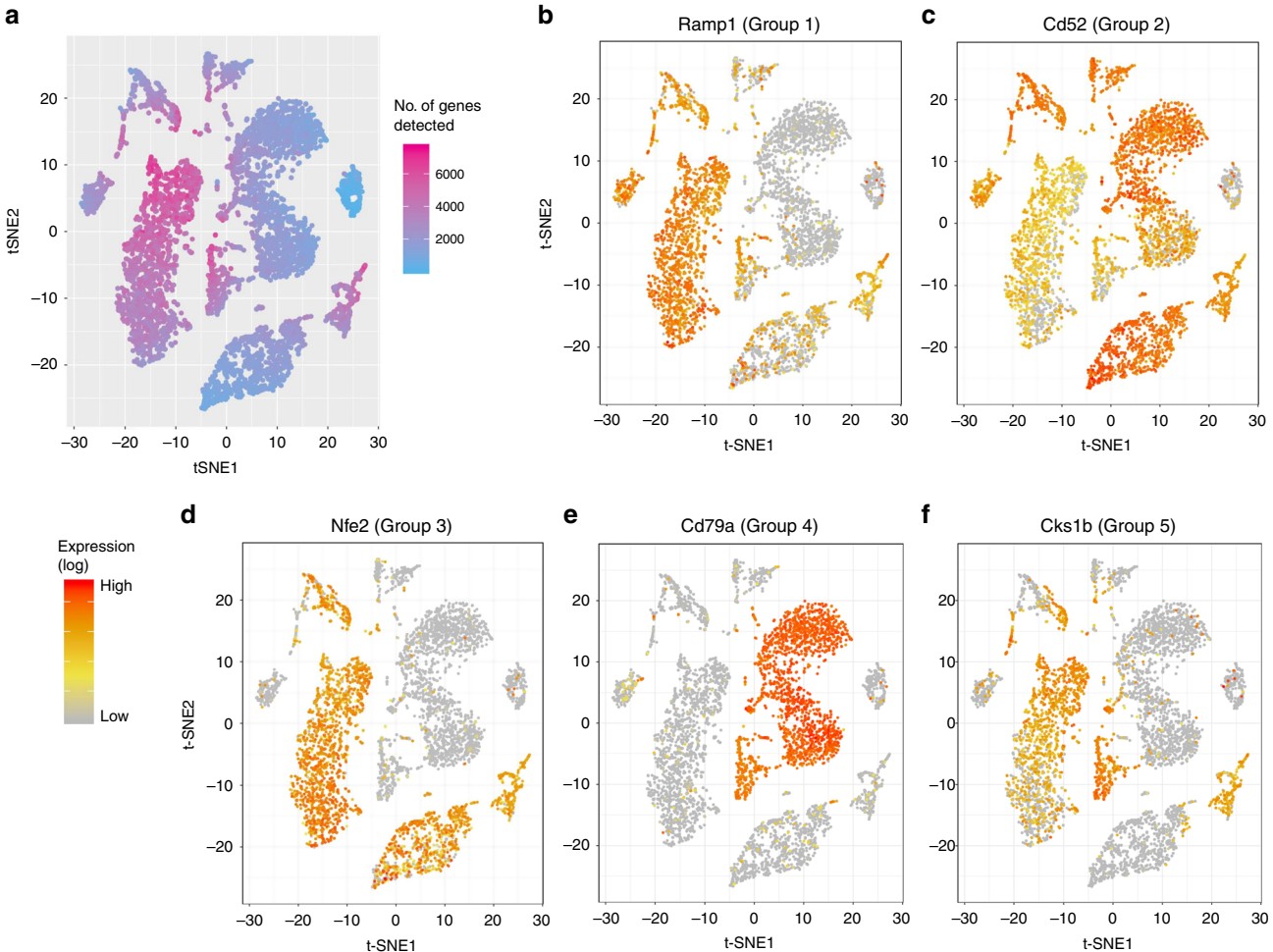

**Fig. 3 Application of singleCellHaystack on bone marrow tissue dataset. a** t-SNE plot of the 5250 cells. The color scale shows the number of genes detected in each cell. **b–f** Expression patterns of five high-scoring DEGs, representative of the five groups in which the genes were clustered.

interpreting singleCellHaystack $p$-values in the conventional definition, the ranking of genes is more relevant.

As an illustration of the usage of singleCellHaystack, we here present results of three example datasets based on different sequencing technologies. In all three cases, the coordinates of cells in the first 50 PCs were used as input, along with the detection levels of all genes in all cells.

Figure 3 summarizes the result of the Tabula Muris bone marrow tissue dataset (FACS-sorted cells). Five-thousand two-hundred fifty cells and 13,756 genes were used as input, and singleCellHaystack took 225 s in the default mode. The t-SNE plot shows a typical mixture of clearly separated as well as loosely connected groups of cells, with considerable variety in the total number of detected genes (Fig. 3a). The most significant DEG was *Ramp1*, which is detected only in a subset of cells (Fig. 3b). To illustrate the variety in expression patterns, we grouped DEGs into five clusters based on hierarchical clustering of their expression in the 50 PC input space (Supplementary Fig. 4). Figure 3c–f show the most significant DEGs of the other 4 groups. Results for two other example datasets are shown in Supplementary Figs. 5 (trachea) and 6 (testis).

To evaluate the dependency of singleCellHaystack on input coordinates and hyperparameter settings (see also discussion in Supplementary Notes), we applied our method using (1) different input spaces (Supplementary Fig. 7A), (2) different bandwidths (Supplementary Fig. 7B), (3) different numbers of grid points (Supplementary Fig. 7C), and different coordinates of

grid points (Supplementary Fig. 7D). We focused here on the three datasets shown in Fig. 3 (bone marrow tissue), Supplementary Fig. 5 (trachea) and Supplementary Fig. 6 (testis). As a reference, we also checked the dependency of a clustering-based method on the number of clusters predicted in a dataset (Supplementary Fig. 7E). Compared with the dependency of cluster-based approaches on the number of predicted clusters, singleCellHaystack is relatively stable w.r.t. bandwidth and the number and coordinates of grid points. In general, similar input spaces (e.g., t-SNE vs. UMAP, or 5 PCs vs. 10 PCs) resulted in similar top-scoring DEGs. However, different input spaces (e.g., 5 PCs vs. 50 PCs) did lead to larger discrepancies in some datasets. For example, the first 50 PCs of the Testis 1 dataset returned different top-scoring DEGs compared to other input spaces (Supplementary Fig. 7A; bottom). For more complex datasets a lower-dimensional input space might not be able to sufficiently capture the variance of the dataset, and a higher dimensional input space might be preferable.

**Comparison with Seurat's** `FindAllMarkers` **function.** Here, we compare our method with the default test used in Seurat's `FindAllMarkers` function (i.e., the Wilcoxon rank-sum test), arguably the most widely used approach. As a representative case, we show the comparison between singleCellHaystack and `FindAllMarkers` on the Tabula Muris bone marrow dataset (Fig. 4). In general, the agreement between both methods was low

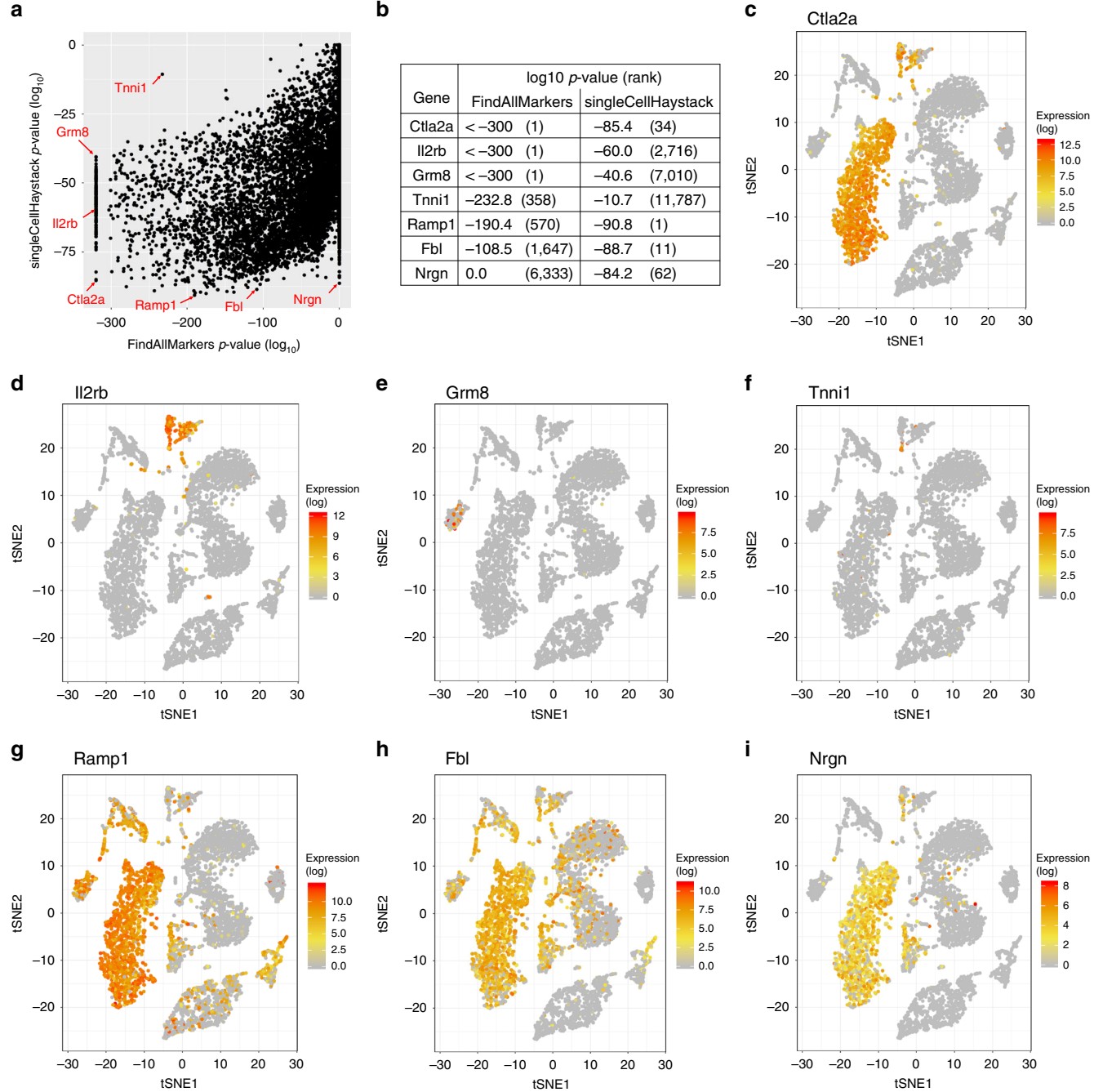

**Fig. 4 Comparison between singleCellHaystack and Seurat.** A comparison between singleCellHaystack and Seurat's `FindAllMarkers` function applied on the Tabula Muris bone marrow tissue dataset. **a** Scatterplot of the *p*-values estimated by `FindAllMarkers` (*x*-axis) and singleCellHaystack (*y*-axis) for all 13,756 genes in the dataset. One-hundred seventy-six genes were given a *p*-value of 0 by `FindAllMarkers`, and are shown as *p*-value 1e-320. Genes highlighted in **c–i** are indicated. **b** Summary of selected genes indicated in **a** are shown. **c–i** Expression patterns of indicated genes.

(Fig. 4a, b and Supplementary Fig. 8). The top 100 high-scoring genes of both approaches have only 3 genes in common (Supplementary Fig. 8). To gain understanding into the difference between a clustering-based approach (`FindAllMarkers`) and the clustering-independent singleCellHaystack, we focus on seven example genes. For the gene *Ctla2a* (Fig. 4c) both methods are in agreement; *Ctla2a* has very high expression in a subset of cells and is not detected in most other subsets.

Some genes are judged to have significant differential expression by `FindAllMarkers` but less so by singleCellHaystack. One such gene is *Il2rb* (Fig. 4d). The expression of this gene closely fits one of the clusters that were used by `FindAllMarkers` to predict DEGs (Supplementary Fig. 9). This trend continues with *Grm8* and *Tnni1*, which have high expression in a handful of cells within a single cluster (Fig. 4e, f and Supplementary Fig. 9).

On the other hand, other genes are picked up by singleCell-Haystack, but are not among the top-ranking genes according to `FindAllMarkers` (Fig. 4g–i). The most significant DEG according to singleCellHaystack is *Ramp1* (Fig. 4g, also shown in Fig. 3b). This gene is expressed across roughly half of the clusters as decided by `FindClusters` (Supplementary Fig. 9), lowering its significance: *Ramp1* is ranked 570th while *Tnni1* is ranked 358th according to `FindAllMarkers`. A similar trend

continues with *Fbl* and *Nrgn*, which have clear differential expression patterns but are not among the top-scoring genes of `FindAllMarkers`.

These representative examples show that clustering-based approaches are likely to overestimate the significance of DEGs whose expression pattern fits very closely with a single cluster (ex: *Tnni*). These approaches are likely to miss DEGs whose expression is spread out over several clusters. On the other hand, singleCellHaystack can detect any pattern of differential expression, independently of the clustering of cells. However, DEGs that are expressed in only a small number of cells (ex: *Grm8*) might be missed.

**Top-scoring DEGs are often known cell type marker genes**. Focusing on results of `haystack_highD` applied on the first 50 PCs of each real dataset, we investigated whether top-scoring DEGs are often known cell type marker genes. For each dataset, we ranked genes by their *p*-value, and counted how often the genes at each rank were high-confidence markers, low-confidence markers, or non-marker genes (see "Methods"). High-confidence marker genes (such as *Cd45* and *Kit*) were strongly enriched among top-scoring DEGs: although they comprise only 2.2% of all genes, on average 32.4% of the top 50 ranked genes were high-confidence cell type markers (Fig. 5).

**The advanced mode considers general gene detection levels**. In scRNA-seq data there can be considerable variation in the number of detected genes between cells. In some datasets this results in clusters of cells with higher or lower general detection levels. The advanced mode of singleCellHaystack can be used to find genes that have expression distributions that are contrary to the general pattern of detected genes (see Methods section). Figure 6 shows three examples, comparing the advanced mode with the default mode. The top-scoring DEGs in the advanced mode are often expressed in cells that have in general fewer detected genes.

**Application on spatial transcriptomics data**. To show that singleCellHaystack can identify DEGs in spatial transcriptomics data we used publicly available data from the 10x Visium platform. We used the mouse brain anterior1 slice dataset, which contains gene expression information for 2696 spots (each corresponding to several cells) together with their 2D spatial coordinates within the tissue. We ran `haystack_2D` using the gene expression and spatial coordinates data. Figure 7 shows the expression of the 6 top-scoring genes returned by `haystack_2D`. The expression distribution of these genes is correlated with brain structures: *Gpr88*, *Pde10a*, and *Rgs9* are expressed in the caudate putamen; *Slc17a7* and *Nptxr* are mainly restricted to the cerebral cortex; *Pcp4l1* is expressed in the olfactory bulb and to a lesser extent in the caudate putamen. This shows that singleCellHaystack can be used on spatial transcriptomics data to identify spatially regulated genes.

**Discussion**

singleCellHaystack is a generally applicable method for finding genes with non-uniform expression distributions in multi-dimensional spaces. Here, we have focused on single-cell transcriptome data analysis, but our approach is also applicable on large numbers of bulk assay samples. We also demonstrated an application on spatial transcriptomics data using the spatial coordinates of cells as input space. singleCellHaystack does not rely on clustering, thus, avoiding biases caused by the arbitrary clustering of cells. It can detect any non-random pattern of expression and can be a useful tool for finding new marker genes. The singleCellHaystack R package includes additional functions for clustering and visualization of genes.

As noted in the Results section, singleCellHaystack found large numbers of statistically significant DEGs in all datasets. The same was true for many of the clustering-based DEG prediction methods. DEG prediction methods often return inflated *p*-values because of the double use of gene expression data (for defining clusters and for DEG prediction)[13,14]. singleCellHaystack suffers from the same issue, because the input coordinates (PCs, t-SNE, or UMAP coordinates) are dimensions containing a large

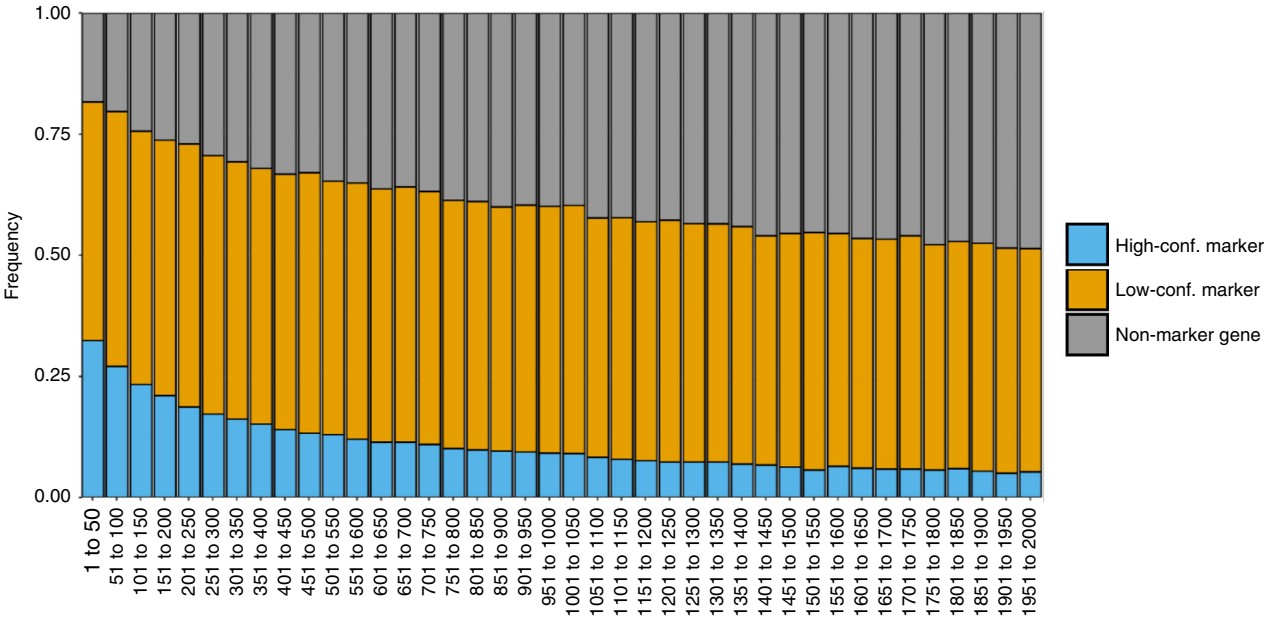

**Fig. 5 Frequencies of cell type marker genes among DEGs.** The frequencies of high-confidence (blue), low-confidence (orange), and non-marker genes (gray) among predicted DEGs in the 136 single-cell datasets are shown. The *x*-axis shows genes ranked by increasing *p*-value in bins of 50 (ranks 1 to 50, ranks 51 to 100, …, up to rank 2000).

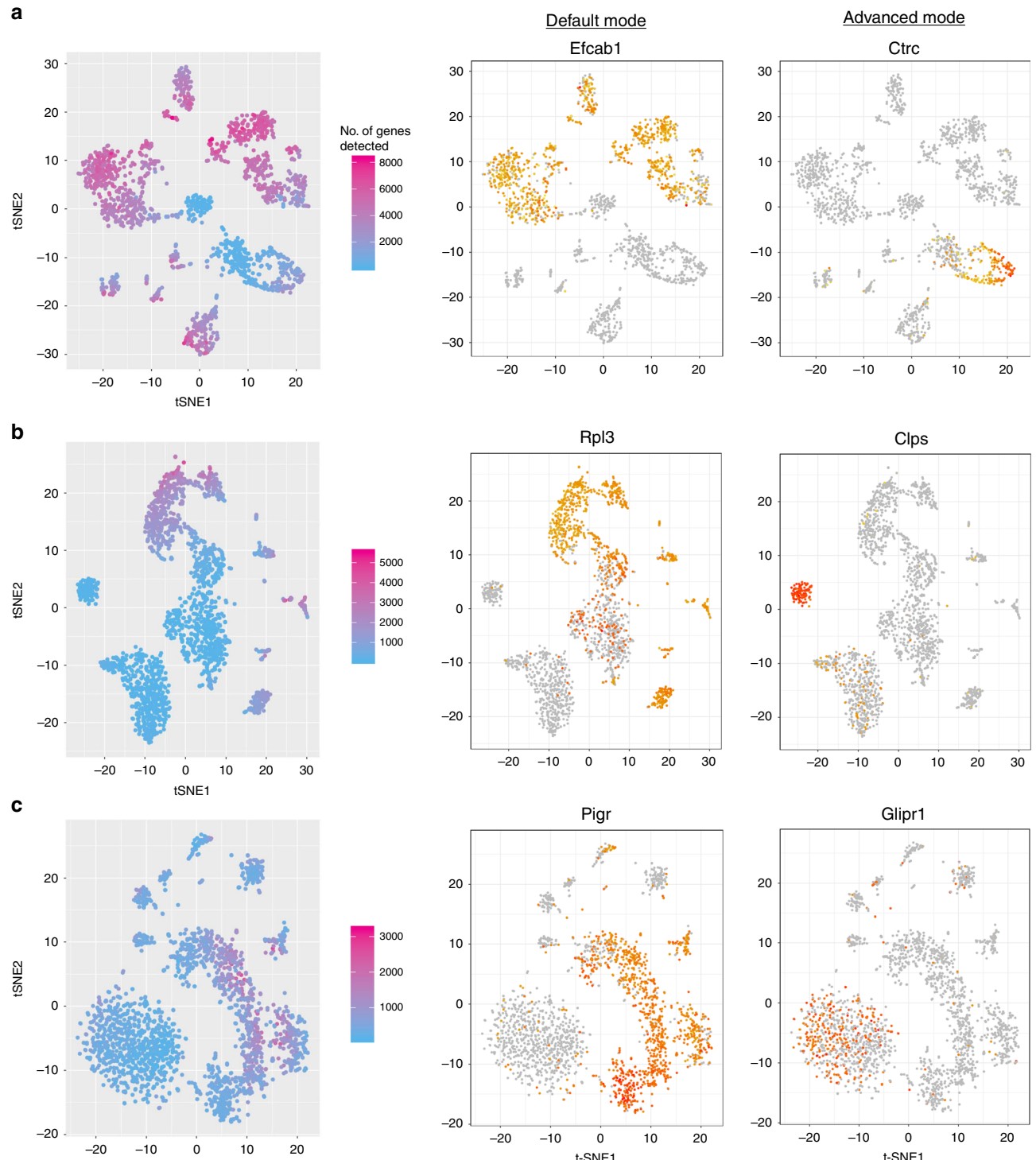

**Fig. 6 Example results of default versus advanced mode of singleCellHaystack.** A t-SNE plot (left), the expression of top-scoring DEGs in the default mode (center) and advanced mode (right) are shown for **a** the Tabula Muris pancreas (FACS-sorted data), **b** the Tabula Muris lung (P8 12; Microfluidic droplet), and **c** the Mouse Cell Atlas small intestine 2 dataset. The color scale for gene expression (center and right panels) is as in Fig. 3.

proportion of the variability in the original data. In future updates we hope to address this issue.

The current implementation of our method requires binary detection data as input (i.e., a gene is either detected or not in each cell). The definition of detection is left to the user and could be modified according to the characteristics of each dataset. When counts >0 is used to define detection, it should be taken into consideration that many zero counts in single-cell data

reflect dropout events due to lack of sensitivity in single-cell techniques. This issue could be partly addressed by applying imputation approaches. However, we are aware that the use of a hard threshold for detection is a weak point of our method. For example, two genes might be detected in the same subset of cells, but one might have ten-fold higher read counts than the other. Since the current implementation of our method is using a hard threshold for detection, it would give both genes the same $D_{KL}$

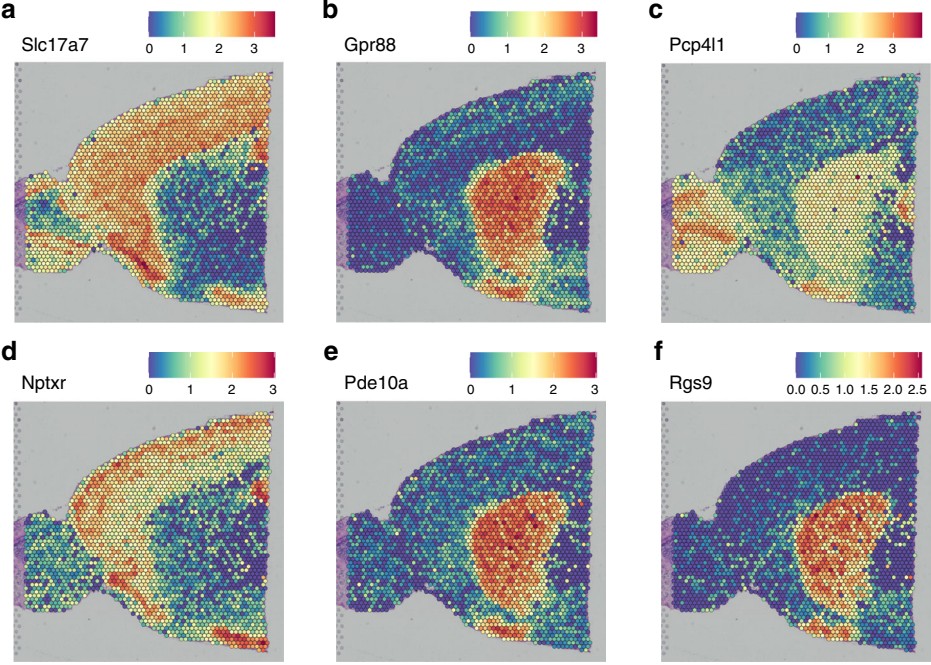

**Fig. 7 Application of singleCellHaystack on spatial transcriptomics data. a–f** Expression levels (normalized counts) per bead in the anterior1 slice of the mouse brain are shown for the top 6 high-scoring genes returned by `haystack_2D`. Each figure shows circles representing the 2696 beads superimposed on a slice of the anterior mouse brain. The locations of the circles correspond to the 2D coordinates of the beads and their colors reflect the expression of each gene.

and *p*-value. In future updates, we will explore ways for using continuous measures of gene expression (such as normalized read counts) as a basis for estimating expression distributions, instead of using binary detection data.

## Methods

**singleCellHaystack methodology**. singleCellHaystack uses the distribution of cells in an input space to predict DEGs. First, it infers a reference distribution of cells in the space (distribution Q). It does so by estimating the local density of cells around a set of fixed grid points in the space using a Gaussian kernel. For 2D spaces (such as t-SNE or UMAP plots, and 2D spatial transcriptomics data), `haystack_2D` divides the 2D space into a grid along both axes, and the intersection points serve as grid points. For multidimensional spaces (such as the first several PCs), `haystack_highD` defines a set of grid points covering the subspace in which the cells are located (see Supplementary Methods).

Next, singleCellHaystack estimates the distribution of cells in which a gene $G$ is detected (distribution $P(G = T)$) and not detected (distribution $P(G = F)$). It does this using the same grid points and Gaussian kernel as used for estimating $Q$. Each distribution is normalized to sum to 1. The definition of detection could depend on the characteristics of the dataset. For the results presented in this paper we used the median read count of each gene as the threshold for defining detection, except for the analysis of the spatial transcriptomics. Alternatively, genes with read counts >0 or exceeding an otherwise defined threshold could be regarded as detected. As an example, we used a fixed threshold of 1 read for the spatial transcriptomics data analysis.

The divergence of gene $G$, $D_{KL}(G)$, is calculated as follows:

$$D_{KL}(G) = \sum_{s \in \{T,F\}} \sum_{x \in \text{grid points}} P(G = s, x) \log \left( \frac{P(G = s, x)}{Q(x)} \right) \quad (1)$$

where $P(G = s, x)$ and $Q(x)$ are the values of $P(G = s)$ and $Q$ at grid point $x$, respectively.

Finally, the significance of $D_{KL}(G)$ is evaluated using randomizations, in which the expression levels of $G$ are randomly shuffled over all cells. The mean and standard deviation of $D_{KL}(G)$ in randomized datasets follow a clear pattern in function of the number of cells in which a gene was detected (see Supplementary Fig. 10 for examples), which is modeled using B-splines[22]. *p*-values are calculated by comparing the observed $D_{KL}(G)$ to the predicted mean and standard deviation (log values).

**singleCellHaystack advanced options**. The distribution $Q$ and the randomizations described above ignore the fact that some cells have in total more detected genes than others. singleCellHaystack can be run in an advanced mode, in which

both the calculation of $Q$ and the randomizations are done by weighting cells by their number of detected genes (see Supplementary Methods for more details).

In addition, singleCellHaystack includes functions for visualization and for clustering genes by their expression distribution in the multidimensional space.

**Application of DEG prediction methods on artificial datasets**. We used Splatter to generate artificial datasets in a range of sizes and complexities, as follows[17]. First, we made datasets containing 1000 cells in 2 clusters; 2000 cells in 3 clusters; 3000 cells in 4 clusters; 4000 cells in 5 clusters; 5000 cells in 6 clusters; 6000 cells in 8 clusters; 7000 cells in 10 clusters; 8000 cells in 15 clusters; 9000 cells in 20 clusters; or 10,000 cells in 25 clusters. For each of these ten settings we generated ten datasets. Secondly, for the same settings of clusters and cell counts we also generated ten datasets each containing randomly determined paths between the clusters (parameter path.from in Splatter's `splatSimulate` function). This gave us a total of 200 datasets (100 with clusters and 100 with paths).

The output of Splatter contains differential expression factors (DEFac[Group]), showing whether a gene has differential expression (factor different from 1) or not (factor = 1) in each group (=cluster). We defined a differential expression score for each gene as the sum of the absolute values of these $\log_2$-transformed factors. We regarded the 100 genes with the highest score as true DEGs.

On these artificial datasets we applied DEsingle, EMDomics, scDD, edgeR, monocle 2, MAST, and the `FindAllMarkers` function of Seurat (see Supplementary Table 1)[7,8,12,18–21]. For a brief description of these methods we refer to Wang et al.[14]. We also tried to apply SCDE and DESeq2 but these had excessively long runtimes even on the smallest datasets, and were, therefore, excluded from this comparison[11,23]. The cell clusters given as input to the DEG prediction methods were detected by running the `FindNeighbors` function of Seurat using the first 15 PCs of each dataset as input followed by the `FindClusters` function using the default resolution and the Smart Local Moving (SLM) algorithm. The same cell clusters were used for all methods. For methods that are implemented in Seurat we used the `FindAllMarkers` function to predict DEGs by comparing each cluster versus all other clusters. For other methods, we implemented this comprehensive comparison (each cluster versus all others). Furthermore, for methods that are implemented in Seurat, we ran `FindAllMarkers` with default options, except for options only.pos = FALSE and return.thresh = 1, and with and without the default filtering (default options logfc.threshold = 0.25 and min.pct = 0.1).

All evaluated methods assign *p*-values and/or scores to genes reflecting their degree of differential expression. We turned these *p*-values and scores into a ranking of genes, from more significant DEGs to less significant DEGs. The ranking was based on *p*-values where possible, using scores to break ties. The ROC method in Seurat does not return *p*-values, so we based its ranking of genes on the predictive power score returned by this method.

Finally, for each DEG prediction method, we calculated the AUC under the ROC curve for every artificial dataset using the ROCR package (version 1.0-7) in R[24].

**scRNA-seq datasets and processing**. For a detailed description we refer to Supplementary Methods. In brief, we downloaded processed data (read counts or unique molecular identifiers) of the Tabula Muris project (FACS-sorted cells: 20 sets; Microfluidic droplets: 28 sets), the Mouse Cell Atlas (Microwell-seq: 87 sets) and a dataset of several hematopoietic progenitor cell types[5,25,26]. PCA was conducted using the 1000 most variable genes. Subsequently, the first 50 PCs were used as input for t-SNE and UMAP analysis, following the recommendations by Kobak and Berens[27]. Finally, singleCellHaystack was run on both 2D (2D t-SNE and UMAP coordinates) and multidimensional (50 PCs) representations of each dataset to predict DEGs. For analyzing runtimes and dependency on input parameters we ran additional runs using 5, 10, 15, and 25 PCs.

**Known cell type marker genes**. We downloaded marker gene data from the CellMarker database[28]. A total of 7852 unique mouse genes are reported as markers in CellMarker, which we split into 630 high-confidence markers (reported in ≥5 publications), and 7222 low-confidence markers (reported in one to four publications). Other genes we regarded as non-marker genes.

**Analysis of dependency on parameters**. We ran singleCellHaystack using different parameter settings as follows:

1. with default parameter settings on different input spaces: t-SNE, UMAP, and the first 5, 10, or 50 PCs.
2. with the default number of grid points on the 50 PC input space using different bandwidths: bandwidth $h$ (default), $2 \times h$, $1.5 \times h$, $h/1.5$, and $h/2$.
3. with the default bandwidth on the 50 PC input space using different number of grid points: 25, 50, 100 (default), 150, and 200.
4. with the default bandwidth on the 50 PC input space with 100 grid points but using 5 different seed values for R's random number generator. This results in different coordinates for the grid points.

The Spearman correlation of $p$-values was used to evaluate the consistency of results of different runs. We also ran Seurat's `FindAllMarkers` function with the default DEG prediction method (Wilcoxon rank-sum test), changing the number of clusters detected in each dataset. First, we ran Seurat's `FindClusters` function with the default resolution parameter setting to obtain the default clusters. Then, we changed the resolution parameter to obtain more (default+1 and +2) and fewer (default-1 and -2) clusters. Finally, we applied the `FindAllMarkers` function on these five different clustering results.

**Analysis of spatial transcriptomics data**. We used mouse brain data from the 10x Genomics Visium spatial transcriptomics platform (https://support.10xgenomics.com/spatial-gene-expression/datasets). We obtained the anterior1 slice with the SeuratData R package for this analysis[29]. The anterior1 slice consists of 31,053 genes and 2696 beads distributed in a 2D lattice. We used Seurat to normalize, obtain the 2D coordinates, and plot the spatial images. Genes with <10 beads with raw counts >1 were removed, resulting in 12,382 genes passing filtering. Raw counts were normalized with the `NormalizeData` function. Normalized counts and spatial coordinates were passed to the function `haystack_2D` to predict spatially differentially expressed genes, using a detection cutoff of 1.

**Reporting summary**. Further information on research design is available in the Nature Research Reporting Summary linked to this article.

## Data availability

The artificial single-cell datasets generated using Splatter for the comparison of singleCellHaystack with other DEG prediction methods are available in figshare with the identifiers 10.6084/m9.figshare.12319787 and 10.6084/m9.figshare.12319247. Other single-cell RNA-seq datasets analyzed in this study are available from https://tabula-muris.ds.czbiohub.org/ (Tabula Muris), https://figshare.com/articles/MCA_DGE_Data/5435866 (Mouse Cell Atlas), GEO accession number GSE81682, and https://support.10xgenomics.com/spatial-gene-expression/datasets (10x Genomics spatial gene expression). The cell type marker data was obtained from the CellMarker database and is available from http://biocc.hrbmu.edu.cn/CellMarker/.

## Code availability

singleCellHaystack is implemented as an R package, available from CRAN (https://CRAN.R-project.org/package=singleCellHaystack) and GitHub (https://github.com/alexisvdb/singleCellHaystack). The repository includes additional instructions for installation in R and example applications.

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

## Acknowledgements

We thank Prof. Yoshio Koyanagi and the members of the Lab. of Systems Virology (Kyoto University), the Lab. of Functional Analysis in silico (Tokyo University), Dr. Yutaro Kumagai (National Institute of Advanced Industrial Science and Technology) and Prof. Wataru Fujibuchi (Kyoto University) for helpful discussions and advice, Tianyu Wang (University of Connecticut) for assistance with DEG prediction scripts in R, and Prof. Daron Standley (Osaka University) and John Rozewicki (Osaka University) for access to the Sysimm computer cluster. This work was supported by an Office of Directors' Research Grant provided by the Institute for Frontier Life and Medical Sciences (Kyoto University), and by the Future Development Funding Program of the Kyoto University Research Coordination Alliance.

## Author contributions

A.V. conceived of the project and methodology. A.V. and D.D. implemented the methods, ran the analyses and wrote the manuscript.

## Competing interests

The authors declare no competing interests.
