## [Peer Review File · Nature Communications]

Reviewers' Comments:

Reviewer #1:

Remarks to the Author:

The authors present a new approach that enables the identification of differentially expressed genes (DEGs) without relying on explicit clustering of cells.

The basic approach proposed in the paper is conceptually appealing. The strategy appears at least plausible as a way of solving the problem posed. However I see three major weakness of the work.

The first is the lack of sufficient methodological details in terms of dependency on the parameter choice (i.e. using t-SNE, PCA or UMAP in the first step, standardizing or not the data, bandwidth h). There is some comparison between the use of PCA vs, t-SNE and UMAP in the supplementary but to me the message conveyed by Figure 9 is a bit unclear. The authors state that there was a high consistency in the ranking of genes, but to me they look different (but maybe I misunderstood the figure meaning) and they depict the mean across 10 runs.

Also some generic statement like "Randomized genes are made that are expressed in cells. This is repeated many times" should be avoided. How much is many?

The second is the lack of sufficient validation. The simulated dataset seems far more simple than real case scenarios. It should be more realistic and, in addition, multiple simulated datasets should be used to assess statistical significance of the results. I would also like to have a comparison with different (more than 2) DE methods on, as stated before, realistic simulated datasets.

The third weakness I see is the paper organization and the use of many generic statements that probably jeopardize the method and results interpretability.

The authors present a new approach that enables the identification of differentially expressed genes (DEGs) without relying on explicit clustering of cells. Exactly for this reason, I would avoid too much focus (both in the abstract and in the introduction) on the clustering task (I found it somehow misleading). The statement that current DE methods are based on clustering and clustering is an arbitrary step is quite generic and need to be motivated.

In general I would like to see

- A more comprehensive description of DE methods (not all of them are based on clustering) and some quantitative statement about their performance.
- The reason and rationale of the proposed methodology
- The general framework with motivation about the choice
- The design of assessment methods (simulation and scoring metrics)
- A brief description of the DE methods you choose to compare with and why (e.g. they are representative of different approaches)
- Etc...

Reviewer #2:

Remarks to the Author:

The manuscript introduces a method to identify differentially expressed genes across unlabeled cells using single cell RNA sequencing data. The proposed method, first, uses conventional dimension reduction methods to represent cells in a low dimensional space using read count data. Then, the method collects some points, called grid points, such that they spread almost uniformly across the low dimensional space. Next, using the low dimensional features of read count values across a cell, it assigns a density distribution to each cell using a Gaussian kernel, in function of their distance to the grid points. A distribution that indicates a gene is detected (or not detected) across the cells is estimated by summing the assigned distribution values over the subsets of cells

in which the gene is detected (or not detected). Finally the method uses Kullback-Leibler Divergence distance to compute the distance between the distribution when a gene is detected (or not detected) and a reference distribution (obtained by summation of the assigned distribution values of all cells). The performance of the proposed method is compared with that of another tool that is designed for identifying differentially expressed genes using single cell sequencing data. Simulated data and three real datasets are used to evaluate the performance of the method.

This work presents a new approach, and identifying differentially expressed genes using unlabeled single cell sequencing data is important to the community. However, the proposed method is not presented convincingly, as explained below.

The density contribution of each cell to each grid point in a low dimensional space is estimated by using a Gaussian kernel, in function of the distance of the cell to the grid point. And, the total density of a gene across cells, $P(G=s)$ (called cell density distribution by the authors, which is confusing), is computed by summing these density contributions. The authors need to justify the use of this density measurement and provide some background information. I am not convinced that this estimation of density can accurately represent the density of a gene (detected or not detected) across cells.

Density estimation is based on the distance of the cells to the grid points in a low dimensional space. This suggests that the results should depend on the coordinate and the number of grid points. However, the detection of grid points is subjective. How robust is the method in regard to grid points at a low dimensional space? What is the effect of changing the number and coordinates of grid points (fixing the low dimension space)? why 100 grid points?

The relationship between read count data across a cell is highly nonlinear. I am not convinced that linear dimension reduction such as PCA can capture this nonlinearity.

In this study zero count is considered as no expression. However, in single cell sequencing data many zero counts are due to dropout event. The authors need to discuss the effect of the dropout event on their methods to give some guidance to the users.

The results of the proposed method are compared to those of only one other method. The authors need to use more tools for the comparison.

The output of the method for a gene is the distance between the distribution of a subset of cells (for detected and not detected genes) and a reference distribution. The distribution of a subset of cells is based on the distance of the cells to the grid points. For genes that are detected in the same subset of cells but have completely different read count values (for example, over the same subset of cells, one gene has the same read count values and another gene has varied read count values, or one gene has very low read count values and another has very high read count values) the method will output the same D_{KL} value. This is because the same subset of cells have the same density distributions, which is the sum of the distances between the subset of cells and the grid points. The authors need to discuss this issue.

Minor comments:

Section 3.5.

Figure S9 is for using different number of PCs and different dimension reduction method. It does not show the performance respect to the different grid points when the same low dimensional space is used.

Line 56. Instead of "multi-dimensional space", "low dimensional space" has to be used. All the analysis are based on the reduced-dimension data.

Line 58. The distributions are computed using a Gaussian kernel and distances to the grid points in a low dimensional space not in the "input space".

Lines 72-75. What does the distribution of cells mean? The authors need to provide a clear definition of the cell distribution in the body of the manuscript so readers can understand it without reading the supplementary materials.

Line 80. This line is confusing. A Gaussian kernel is used to estimate the contributions of density of cells respect to the grid points, not " at grid points".

Line 82. How is a "detected" gene defined? Is a threshold of zero is used for read count values? The authors need to clarify it.

line 112. What does "genes with biased expression patterns" mean? Does it mean differentially expressed genes?

line 193. In section 2.3 it is mentioned that 1000 most variable genes were selected. But in this line it is mentioned that more that 13,000 genes were used. Please clarify it.

Section 2.4. In the previous section, it is mentioned that only 1000 gens are deleted for the analysis. In this section 7852 marker genes are used. The author need to clarify how many of these marker genes are included in this study.

The fonts of some figures, specially in the supplementary materials, are too small and not readable.

Response to all Reviewers

We thank the reviewers for their constructive comments. Below, please find our point-by-point responses to all comments and questions. We believe that we have examined each of the points raised and could respond thoroughly. As suggested, we have made extensive additions and changes to the manuscript. Again, we would like to thank both reviewers for being interested in our manuscript and for their thoughtful suggestions, which have definitely made this a stronger paper.

List of the main changes:

1. Explained underlying concept more clearly (new section 3.1 and new Fig. 1).
2. More extensive comparison with existing DEG prediction approaches applied on 200 artificial datasets of varying complexity (new sections 2.3 and 3.2, new figure S2 and S3, and Tables S1-3). We removed the original toy artificial dataset example, which was too simple.
3. Analysis of dependency on parameters (new section 2.6, last paragraph of 3.3 and new Supplementary section “3 Dependency on parameters” and figure S8). We also removed the original Fig. S9, which was hard to understand, and the corresponding part of the manuscript.
4. A new example application on a spatial transcriptomics dataset (new sections 2.7 and 3.7, and new Figure 7).
5. A new paragraph in Conclusions about the use of a hard threshold for defining “detection” in our approach.
6. Improved quality of figures (increased font size and resolution where needed)

Specific answers to the reviewers’ comments are shown below. Our responses are marked in green. Changes in the manuscript have been indicated in red.

Reviewers' comments:

Reviewer #1 (Remarks to the Author):

The authors present a new approach that enables the identification of differentially expressed genes (DEGs) without relying on explicit clustering of cells.

The basic approach proposed in the paper is conceptually appealing. The strategy appears at least plausible as a way of solving the problem posed. However I see three major weakness of the work.

The first is the lack of sufficient methodological details in terms of dependency on the parameter choice (i.e. using t-SNE, PCA or UMAP in the first step, standardizing or not the data, bandwidth h). There is some comparison between the use of PCA vs, t-SNE and UMAP in the supplementary but to me the message conveyed by Figure 9 is a bit unclear. The authors state that there was a high consistency in the ranking of genes, but to me they look different (but maybe I misunderstood the figure meaning) and they depict the mean across 10 runs.

Also some generic statement like “Randomized genes are made that are expressed in cells. This is

repeated many times” should be avoided. How much is many?

Thank you for this feedback. We realize that the original Figure S9 was hard to understand. We have removed it and the corresponding part of the paper. Instead, we have added a new section (see “3 Dependency on parameters” in Supplement and new Figure S8) in which we analyzed the dependency of our results on 1) the input space, and a few hyperparameters: 2) the bandwidth, 3) the number of grid points, and 4) the coordinates of the grid points. For comparison, we also looked at how a cluster-based approach depends on the number of clusters predicted in each dataset.

Naturally, results did change depending on the input space, which is to be expected (the results of other methods also change with the input space). We should not expect the 2D t-SNE or UMAP space to fully capture the structure of the data in the first 50 PC dimensions. But our method was in general consistent when using different bandwidths or numbers of grid points: top-scoring DEGs remain top-scoring DEGs regardless of the hyperparameter values. On the other hand, the results of a cluster-based approach (we used Seurat’s default method as a representative example here) changed more drastically when the number of clusters changed (Fig. S8E).

Regarding the generic statements, we have rewritten the part mentioned (see “Step 5: Estimating the significance of $D_{KL}(G)$ ” in supplement) and made similar improvements to the text in other places. We also made changes to the consistency of terms like “expression distribution” and avoided terms like “biased expression” (which we replaced by “differential expression”).

The second is the lack of sufficient validation. The simulated dataset seems far more simple than real case scenarios. It should be more realistic and, in addition, multiple simulated datasets should be used to assess statistical significance of the results. I would also like to have a comparison with different (more than 2) DE methods on, as stated before, realistic simulated datasets.

We agree that the original toy dataset was too simple. We have replaced it by an extensive analysis of 200 artificial datasets (generated using Splatter), with varying levels of complexity (from dataset of 1,000 cells in 2 groups to dataset of 10,000 cells in 25). These datasets are more relevant to current real datasets. On these datasets we applied our own method as well as 14 other DEG prediction approaches (see Supplementary Table S1), and we compared accuracy (using AUC) and runtimes. In general, singleCellHaystack showed comparatively high accuracy (highest median AUC for datasets of 2,000 or more cells) with shorter runtimes. For more details we refer to Figure 2, and supplementary Tables S2-3, and Fig. S3.

We believe these results clearly show that singleCellHaystack is an attractive method for DEG prediction in single-cell datasets.

The third weakness I see is the paper organization and the use of many generic statements that probably jeopardize the method and results interpretability.

The authors present a new approach that enables the identification of differentially expressed genes (DEGs) without relying on explicit clustering of cells. Exactly for this reason, I would avoid too much focus (both in the abstract and in the introduction) on the clustering task (I found it somehow misleading). The statement that current DE methods are based on clustering and clustering is an arbitrary step is quite generic and need to be motivated.

In general I would like to see

- A more comprehensive description of DE methods (not all of them are based on clustering) and some quantitative statement about their performance.
- The reason and rationale of the proposed methodology
- The general framework with motivation about the choice
- The design of assessment methods (simulation and scoring metrics)
- A brief description of the DE methods you choose to compare with and why (e.g. they are representative of different approaches)
- Etc...

Thank you for this valuable comment. We have made several changes to the paper to make it easier to understand, including the addition of a new section explaining the underlying concept more clearly (section 3.1 and Fig. 1), and an extensive comparison with many other widely used DEG prediction approaches (section 3.2, Figure 2, Supplementary Tables S1, S2, and S3, and Figure S3).

Regarding the focus in clustering, we would like to respectfully disagree, and stress this issue clearly in the Abstract and Introduction. One of our concerns is that we want to avoid presenting our method as “yet another DEG prediction approach”. The key difference of our approach is that – unlike existing approaches – it does not require pre-defined groups of cells to be compared with each other. This is what makes our approach unique. So, we feel we should stress this point quite clearly in the Abstract and the Introduction. We saw this confirmed again while doing our comparison with other approaches: they required pre-defined groups of cells. In fact, several methods accept only 2 groups of cells to compare between, making them even more restricted.

We hope for your understanding.

Reviewer #2 (Remarks to the Author):

The manuscript introduces a method to identify differentially expressed genes across unlabeled cells using single cell RNA sequencing data. The proposed method, first, uses conventional dimension reduction methods to represent cells in a low dimensional space using read count data. Then, the method collects some points, called grid points, such that they spread almost uniformly across the low dimensional space. Next, using the low dimensional features of read count values across a cell, it assigns a density distribution to each cell using a Gaussian kernel, in function of their distance to the grid points. A distribution that indicates a gene is detected (or not detected) across the cells is estimated by summing the assigned distribution values over the subsets of cells in which the gene is detected (or not detected). Finally the method uses Kullback-Leibler Divergence distance to compute the distance between the distribution when a gene is detected (or not detected) and a reference distribution (obtained by summation of the assigned distribution values of all cells). The performance of the proposed method is compared with that of another tool that is designed for identifying differentially expressed genes using single cell sequencing data. Simulated data and three real datasets are used to evaluate the performance of the method.

This work presents a new approach, and identifying differentially expressed genes using unlabeled single

cell sequencing data is important to the community. However, the proposed method is not presented convincingly, as explained below.

The density contribution of each cell to each grid point in a low dimensional space is estimated by using a Gaussian kernel, in function of the distance of the cell to the grid point. And, the total density of a gene across cells, $P(G=s)$ (called cell density distribution by the authors, which is confusing), is computed by summing these density contributions. The authors need to justify the use of this density measurement and provide some background information. I am not convinced that this estimation of density can accurately represent the density of a gene (detected or not detected) across cells.

Thank you for this comment. We have added a new section explaining more clearly the underlying concept of our approach (new section 3.1 and new Fig. 1) using the bone marrow dataset from Tabula muris as example. We show the expression pattern of 4 marker genes, using typical tSNE plots as well as plots showing the grid points used in our approach. The color of the grid points reflects $P(G=T)/Q$. From the plots we can see that grid points are roughly distributed in a way that covers the cells uniformly, and that we can use them to measure the local “density” of cells in which each gene is detected.

We have tried to make the text less confusing. For example, we now use the term “expression distribution” in a more consistent way to describe the distribution of cells expressing a gene in the input space. We also replaced instances of the confusing “biased expression” with more widely used term “differential expression”.

Density estimation is based on the distance of the cells to the grid points in a low dimensional space. This suggests that the results should depend on the coordinate and the number of grid points. However, the detection of grid points is subjective. How robust is the method in regard to grid points at a low dimensional space? What is the effect of changing the number and coordinates of grid points (fixing the low dimension space)? why 100 grid points?

We have added a new section (see “3 Dependency on parameters” in Supplement and new Figure S8) in which we analyzed the dependency of our results on 1) the input space, and a few hyperparameters: 2) the bandwidth, 3) the number of grid points, and 4) the coordinates of grid points. For comparison, we also looked at how a cluster-based approach depends on the number of clusters predicted in each dataset.

With regard to changing the number of grid points, we found that results are relatively robust: top-scoring DEGs remain top-scoring DEGs regardless of the number of grid points. However, we expect that the default 100 grid points might not be sufficient for datasets with a high heterogeneity; i.e. if there are more than 100 distinct types of cells in a dataset, 100 grid points will obviously not be sufficient for covering the entire subspace taken up by the cells. We added a few lines on this in the supplementary section “Dependency of *singleCellHaystack* on the number of grid points”.

We admit that the grid points are subjective. However, in defense of our method, so is the clustering (both the number of clusters and the cell-to-cluster assignments) used by clustering approaches.

The relationship between read count data across a cell is highly nonlinear. I am not convinced that linear dimension reduction such as PCA can capture this nonlinearity.

We agree on this 100%. But dimensionality reduction approaches are beyond the scope of this study. Our method is in principle applicable on any type of coordinates, including those returned by PCA, UMAP, tSNE, or any other approaches, as well as spatial coordinates in spatial transcriptomics.

In this study zero count is considered as no expression. However, in single cell sequencing data many zero counts are due to dropout event. The authors need to discuss the effect of the dropout event on their methods to give some guidance to the users.

We have added some discussion on this in Conclusion.

In addition, we noticed that this was a mistake in our submission. Originally, we did use 0 as a threshold for detection (i.e. counts or UMI > 0). But during the development of the method, we noticed that some genes have > 0 reads in all cells. Therefore, we changed the threshold to the median count of the gene over all cells. This was the threshold which we used in our original submission. Of course, for many genes the median count is 0. We apologize for the confusion. In the future we hope to update our method so that no hard threshold is needed.

The results of the proposed method are compared to those of only one other method. The authors need to use more tools for the comparison.

We have conducted a more extensive comparison using 200 artificial datasets (generated using Splatter), with varying levels of complexity (from datasets with 1,000 cells distributed in 2 clusters to datasets with 10,000 cells distributed in 25 clusters). These datasets are more similar to current real datasets. On these datasets we applied our own method as well as 14 other DEG prediction approaches (see Supplementary Table S1), and we compared accuracy (using AUC) and runtimes. In general, singleCellHaystack showed comparatively high accuracy (highest median AUC for datasets of 2,000 or more cells) with short runtimes. For more details we refer to Figure 2, and supplementary Tables S2-3, and Fig. S3.

We believe these results clearly show that singleCellHaystack is an attractive method for DEG prediction in single-cell datasets.

The output of the method for a gene is the distance between the distribution of a subset of cells (for detected and not detected genes) and a reference distribution. The distribution of a subset of cells is based on the distance of the cells to the grid points. For genes that are detected in the same subset of cells but have completely different read count values (for example, over the same subset of cells, one gene has the same read count values and another gene has varied read count values, or one gene has very low read count values and another has very high read count values) the method will output the same D_{KL} value. This is because the same subset of cells have the same density distributions, which is the sum of the distances between the subset of cells and the grid points. The authors need to discuss this issue.

We realize that this hard threshold and binary distinction is a weak point of our method, even if it has comparatively high accuracy to existing methods. We will explore feasible ways for using normalized

counts directly as input for estimating expression distributions in future updates. We have added some discussion about this weak point in the Conclusions section.

Minor comments:

Section 3.5.

Figure S9 is for using different number of PCs and different dimension reduction method. It does not show the performance respect to the different grid points when the same low dimensional space is used.

We feel that the original Figure S9 was hard to understand (as also pointed out by Reviewer 1). We have removed the figure and the corresponding analysis from the paper. Instead we added a new analysis of dependency on parameters (see also new Fig. S8). This also includes a comparison of results with different seed values for R's random number generator, which leads to different grid point coordinates (Fig. S8D). In general, our method returns similar results even when different grid points are used.

Line 56. Instead of "multi-dimensional space", "low dimensional space" has to be used. All the analysis are based on the reduced-dimension data.

It is true that the analyses are based on reduced-dimensional data, but we would prefer to continue using multidimensional (= "of or involving several dimensions"). We feel that "low dimensional" suggests rather 1~5 dimensions in this field of research, while most of the results we present are based on 50 dimensions. In the line in question, we merely want to say that the input space can be 2 or more dimensions.

Line 58. The distributions are computed using a Gaussian kernel and distances to the grid points in a low dimensional space not in the "input space".

We are sorry, but we failed to understand the problem with the original sentence. We are using "input space" here to mean the space that is given as input to our method. Our method estimates the distribution of the cells in that space. It does so using distances to the grid points, which are also located in that same input space. We hope that as a brief summary in the Introduction this sentence is not wrong or misleading. We also added a new sentence in the first paragraph of the Materials and methods section, giving some examples of input spaces. However, it is possible that we are missing the point of your comment. In that case we would be more than happy to make adjustments as needed.

Lines 72-75. What does the distribution of cells mean? The authors need to provide a clear definition of the cell distribution in the body of the manuscript so readers can understand it without reading the supplementary materials.

We have improved the text to consistently use the term "expression distribution" of a gene, by which we mean the distribution of cells expressing a gene in the input space. We have improved the explanation in section 2.1 and added a new section 3.1 which explains the idea behind our method. We hope this will help readers to understand the concept of our method.

Line 80. This line is confusing. A Gaussian kernel is used to estimate the contributions of density of cells respect to the grid points, not " at grid points".

We have fixed this error.

Line 82. How is a “detected” gene defined? Is a threshold of zero is used for read count values? The authors need to clarify it.

Thank you. As also mentioned above, we use the median read count of each gene as a threshold for defining detection. If the count for a gene in a cell is higher than its median count, we define the gene as detected in that cell. For many (but not all) genes the median count is 0. We have improved the explanation in the text.

line 112. What does “genes with biased expression patterns” mean? Does it mean differentially expressed genes?

Our apologies for using confusing terms. We have changed this to differentially expressed genes, as suggested.

line 193. In section 2.3 it is mentioned that 1000 most variable genes were selected. But in this line it is mentioned that more that 13,000 genes were used. Please clarify it.

We have improved the explanation in the text.

Section 2.4. In the previous section, it is mentioned that only 1000 gens are deleted for the analysis. In this section 7852 marker genes are used. The author need to clarify how many of these marker genes are included in this study.

We have made the section about the CellMarker data clearer.

The fonts of some figures, specially in the supplementary materials, are too small and not readable.

We have improved the quality of the plots paying attention to the font sizes.

Reviewers' Comments:

Reviewer #1:

Remarks to the Author:

I think the authors have done a great job and answered all my comments.

I have two minor comments

A lot of information has been included to evaluate the consistency of results across different method and using different input space.

Spearman correlation of p-values or scatter plots of pvalues were used at this purpose. Still, as user, I would like to have some indication in terms of precision/recall obtained using different input spaces. Does it depends on the dataset? Can you give a general message about this choice?

English need some revision

Reviewer #2:

Remarks to the Author:

Authors have adequately addressed all the comments. I do not have any further major comment. The only minor comment is that the new chapter 3.7 needs more clarification.

Response to all Reviewers

We thank the reviewers for reviewing our manuscript one more time. Below, please find our point-by-point responses to their comments and suggestions. We have made some adjustments accordingly.

List of the main changes:

1. We have added our R package to CRAN, so we added this to the manuscript.
2. We have made some changes to make section “Application on spatial transcriptomics data” (section 3.7) more clear. We removed the original Figure 7A panel (Total counts per bead) because this might have been confusing. Instead of showing the top 5 genes, the new Figure 7 is now showing the top 6 genes. We also improved the explanation in the text and in the legend of Figure 7. Related to that, we noticed that our original Figure 7 showed the top scoring genes WITHOUT normalizing the expression data, while our methods said that the data was normalized first. We updated the figure to show the top-scoring genes after normalization. We apologize for the confusion. Please note that 2 of the original top 5 genes remain in the new top 6, showing that the results are consistent.
3. We made several changes to the format of the manuscript to make it fit the requirements of Nature Communications, including moving the Methods section to the end of the paper.

Specific answers to the reviewers’ comments are shown below. Our responses are marked in green.

Reviewer #1 (Remarks to the Author):

I think the authors have done a great job and answered all my comments.

I have two minor comments

A lot of information has been included to evaluate the consistency of results across different method and using different input space.

Spearman correlation of p-values or scatter plots of pvalues were used at this purpose. Still, as user, I would like to have some indication in terms of precision/recall obtained using different input spaces. Does it depends on the dataset? Can you give a general message about this choice?

Thank you for this comment. Please note that Supplementary figure 7A shows a comparison between different input spaces (t-SNE, UMAP, 5, 10 and 50 PCs). The choice of input space should depend on the expected complexity of the dataset. For complex datasets with many cell types, few PCs, t-SNE or UMAP or a low number of PCs may not be able to capture the full heterogeneity. For example, in Supplementary figure 7A (Testis) we found that running haystack with 50 PCs returned different top-scoring DEGs compared to other input spaces. This shows that for complex datasets higher dimensional input spaces may be preferable.

The original Supplementary section “Dependency of singleCellHaystack on input space” contained some discussion about this. We added a few lines in the main text (last paragraph of section “Application to 136 real single-cell datasets”).

English need some revision

We have checked the manuscript again and made several small corrections.

Reviewer #2 (Remarks to the Author):

Authors have adequately addressed all the comments. I do not have any further major comment. The only minor comment is that the new chapter 3.7 needs more clarification.

Thank you for pointing this out. As we noted above, we have made several adjustments to the text (section “Application on spatial transcriptomics data”, previously section 3.7) as well as to the legend of Figure 7.